# Antarctic Marine Bacteria as a Source of Anti-Biofilm Molecules to Combat ESKAPE Pathogens

**DOI:** 10.3390/antibiotics12101556

**Published:** 2023-10-21

**Authors:** Marco Artini, Rosanna Papa, Gianluca Vrenna, Marika Trecca, Irene Paris, Caterina D’Angelo, Maria Luisa Tutino, Ermenegilda Parrilli, Laura Selan

**Affiliations:** 1Department of Public Health and Infectious Diseases, Sapienza University, p.le Aldo Moro 5, 00185 Rome, Italy; marco.artini@uniroma1.it (M.A.); gianluca.vrenna@opbg.net (G.V.); marika.trecca@uniroma1.it (M.T.); paris.1744509@studenti.uniroma1.it (I.P.); laura.selan@uniroma1.it (L.S.); 2Research Unit of Diagnostical and Management Innovations, Children’s Hospital and Institute Research Bambino Gesù, 00165 Rome, Italy; 3Department of Chemical Sciences, Federico II University, Complesso Universitario Monte S. Angelo, Via Cintia 4, 80126 Naples, Italy; caterina.dangelo@unina.it (C.D.); tutino@unina.it (M.L.T.); erparril@unina.it (E.P.)

**Keywords:** *Enterococcus faecium*, *Staphylococcus aureus*, *Klebsiella pneumoniae*, *Acinetobacter baumannii*, *Pseudomonas aeruginosa*, *Enterobacter* sp., biofilm

## Abstract

The ESKAPE pathogens, including bacteria such as *Enterococcus faecium*, *Staphylococcus aureus*, *Klebsiella pneumoniae*, *Acinetobacter baumannii*, *Pseudomonas aeruginosa*, and *Enterobacter* species, pose a global health threat due to their ability to resist antimicrobial drugs and evade the immune system. These pathogens are responsible for hospital-acquired infections, especially in intensive care units, and contribute to the growing problem of multi-drug resistance. In this study, researchers focused on exploring the potential of Antarctic marine bacteria as a source of anti-biofilm molecules to combat ESKAPE pathogens. Four Antarctic bacterial strains were selected, and their cell-free supernatants were tested against 60 clinical ESKAPE isolates. The results showed that the supernatants did not exhibit antimicrobial activity but effectively prevented biofilm formation and dispersed mature biofilms. This research highlights the promising potential of Antarctic bacteria in producing compounds that can counteract biofilms formed by clinically significant bacterial species. These findings contribute to the development of new strategies for preventing and controlling infections caused by ESKAPE pathogens.

## 1. Introduction

The ESKAPEs are a group of pathogens that pose a global health threat due to their ability to escape the biocidal action of antimicrobials and the immune system response. This group of pathogens is composed of both Gram-positive and Gram-negative bacterial species, namely: *Enterococcus faecium*, *Staphylococcus aureus*, *Klebsiella pneumoniae*, *Acinetobacter baumannii*, *Pseudomonas aeruginosa,* and *Enterobacter* species [1].

ESKAPE pathogens are mainly responsible for nosocomial infections, causing infections that are defined as hospital-acquired infections (HAIs) that affect patients within 48 h of hospital admission, 3 days after discharge, or 30 days after a surgical intervention [2]. Furthermore, ESKAPE pathogens are responsible for more than 40% of infections in intensive care units (ICUs) and require unaffordable economic expenses, especially in developing countries [3]. Over the years, an increasing number of pathogens have been reported as becoming multi-drug resistant (MDR) as a result of the misuse and the abuse of antibiotics [4,5]. A new report from the European Centre for Disease Prevention and Control (ECDC) ascribes over 35,000 deaths a year, and 874,000 disability-adjusted life years, for complications caused by hospital-acquired (HA) and community-acquired (CA) antimicrobial resistance (AMR), accounting for EUR 1.4 billion in direct and indirect costs.

ESKAPE pathogens have developed resistance mechanisms against the major classes of antibiotics such as oxazolidinones, aminoglycosides, macrolides, fluoroquinolones, tetracyclines, and beta-lactam, also in combination with inhibitors, through genetic mutations and the acquisition of mobile genetic elements [6]. Furthermore, they also became resistant to the last line of defense, represented by carbapenems and glycopeptides [7]. These pathogens display drug resistance via numerous and different phenotypic and/or genotypic mechanisms, such as drug inactivation by irreversible enzyme cleavage, drug-binding site alteration, diminution in permeability of drug or drug efflux increment to reduce its accumulation, or by the production of biofilm [8,9]. Overall, the constitutive and/or inducible expression of these drug-resistance mechanisms led to the increased representation of these species in HA infections [10]. Moreover, the heterogenicity of the antimicrobial-resistance profile within the same bacterial species remarkably complicates the development of new effective therapies [5]. Subsequently, it is important to find alternative targets to inhibit the growth and spread of pathogens, taking into account that the emergence of MDR bacteria has been paralleled by a declining antibiotic development pipeline [11].

The scientific community has shown significant interest in using novel strategies to counteract the virulence of MDR pathogens, including ESKAPEs. In this context, the anti-virulence strategy can be used to combat the emergence of antibiotic-resistant pathogens. Anti-virulence drugs do not necessarily kill bacterial cells but prevent bacterial pathogenesis by targeting their virulence traits [12]. In this approach, the anti-ESKAPE drugs would interfere with bacterial virulence factors to treat disease, thus leading to the development of new strategies for the prevention and control of infections [12].

In the search for new anti-virulence drugs, the development of anti-biofilm strategies is therefore of major interest, and currently represents an important field of investigation in which nonbiocidal molecules are highly valuable to avoid the rapid appearance of MDR species [13].

The aim of this paper was to search for new anti-biofilm activities against ESKAPE pathogens. We focused our attention on anti-biofilm molecules since these compounds do not induce the appearance of escape mutants and can be used in combination with conventional antibiotics to increase their activity.

In the research of new anti-biofilm agents, microorganisms able to thrive in harsh conditions, like in Antarctica, represent an unexploited reservoir of biodiversity [14]. Indeed, Antarctic marine bacteria established different survival strategies to live in extreme environmental conditions [15] that decrease the presence of competitive microorganisms. Such behavior is mandatory when nutrients are limited or difficult to uptake. Biofilm formation allows cells to grow even in oligotrophic environments [16,17], and the production of anti-biofilm molecules can reduce the biofilm formation of competitors and their ability to survive. Therefore, it is not surprising that some recent papers report that marine Antarctic bacteria produce and secrete anti-biofilm molecules [13,18,19,20,21,22,23]. Furthermore, cell-free supernatants and organic extracts obtained from different bacterial cultures of Polar marine bacteria showed interesting anti-biofilm activities on *P. aeruginosa* and *S. aureus* [13,19,22].

In this work, we analyzed the effect of supernatants derived from four selected Antarctic marine bacteria, belonging to *Pseudoalteromonas*, *Psychrobacter*, and *Pseudomonas* genera, against ESKAPE pathogens.

The anti-biofilm effects of Antarctic bacterial-cell-free culture supernatants (SNs) were examined on 60 clinical ESKAPE pathogens either during biofilm development, by adding them to the medium at the beginning of the growth, or after biofilm formation. Firstly, their antimicrobial activity was evaluated to exclude any effect of SNs on bacterial viability; the results did not highlight any antimicrobial activity on all ESKAPE pathogens. On the contrary, SNs were able to prevent biofilm formation and promote the disaggregation of mature biofilm.

The obtained results have shown the great potential of Antarctic bacteria as producers of molecules that counteract the biofilm of bacterial species of significant clinical interest.

## 2. Results

### 2.1. Phenotypic Characterization of Clinical and Reference Strains

Clinical strains belonging to ESKAPE pathogens were characterized using their antimicrobial profile following the guidelines reported in the EUCAST Clinical Breakpoint Tables v. 13.0 (valid from 1 January 2023). Results are summarized in Table 1, Table 2, Table 3, Table 4, Table 5 and Table 6.

As expected, all strains had different resistance profiles.

Regarding Enterococcus species, we analyzed nine clinical strains and *E. faecalis* ATCC 29212 as a reference strain (Table 1). The latter was originally isolated from urine, and it is commonly used for antimicrobial susceptibility tests and urinary tract infection research. All bacterial strains are sensitive to tetracyclines and glycopeptides. They showed variable resistance to fluoroquinolones (three out of ten were resistant) and oxazolidones (eight out of ten were resistant), while all strains possessed an intermediate resistance to carbapenems.

Table 2 reports the antibiotic resistance profiles of eight clinical and two ATCC strains of *S. aureus*. *S. aureus,* ATCC 6538P and ATCC 29213 reference strains (originally isolated from a wound), which are commonly used for susceptibility disc testing, quality control, and drug discovery research. As expected, *S. aureus* ATCC 6538P was sensitive to all tested antibiotics, while *S. aureus* ATCC 29213 was resistant to fluoroquinolones and vancomycin. All clinical strains were resistant to fluoroquinolones and vancomycin, and sensitive to amikacin, tetracyclines, and trimethoprim/sulfamethoxazole. Seven out of eight clinical strains are resistant to penicillins and five out of eight also to cephalosporins.

We analyzed ten *K. pneumoniae* strains, including nine clinical strains and ATCC 13883 commonly used for quality control and antimicrobial testing (Table 3). All strains were mostly resistant to penicillins, fluoroquinolones, cephalosporines, and monobactams (except for ATCC 13883 which was sensitive to cephalosporines, amoxicillin/clavulanic acid, and aztreonam). We also observed a variable resistance to imipenem (five out of ten), amikacin (five out of ten), and trimethoprim/sulfamethoxazole (six out of ten).

Regarding *A. baumannii* species, we analyzed eight clinical and two ATCC strains (Table 4). ATCC 17978, originally isolated from a fatal case of meningitis in a 4-month-old human, was susceptible to all classes of tested antibiotics (with an intermediate resistance only for trimethoprim/sulfamethoxazole), while ATCC 19606, firstly isolated from human urine, was resistant to aminoglycosides and trimethoprim/sulfamethoxazole. All clinical strains were multi-drug resistant except strain Ab12 which was sensitive to all tested antibiotics.

The *P. aeruginosa* strains used in this work are listed in Table 5. We used two ATCC strains, both commonly used for standard antimicrobial disk susceptibility tests, and eight clinical strains. All strains were commonly susceptible to all classes of antibiotics, except strain PA26 which was resistant to imipenem and levofloxacin.

For *E. coli* we used ten bacterial strains, including two ATCC strains both reported as reference strains for antibiotic susceptibility testing and sensitive to all classes of antibiotics (Table 6), and eight clinical strains; they showed variable resistance, except strain EC2 which was sensitive to all classes of antibiotics. Only strain EC1 was resistant to amikacin, while strains ECNDM2, ECNDM4, and ECNDM5 were resistant to all tested antibiotics save amikacin.

### 2.2. Selection of Antarctic Marine Bacteria

In this work, we analyzed the effect of supernatants derived from selected Antarctic marine bacteria against ESKAPE pathogens. In detail, we selected *Pseudoalteromonas haloplanktis* TAC125 [24] which produces a secreted molecule, the pentadecanal, able to inhibit the biofilm growth of Staphylococcus epidermidis [18,25]. The second marine strain chosen was *Psychrobacter* sp. TAE2020 [26]; it produces and secretes molecules active against some virulence factors of *P. aeruginosa* isolated from patients affected by cystic fibrosis, such as biofilm formation and accumulation, pyocyanin production, and swimming and swarming motility [22]. *Pseudomonas* sp. TAE6080 cell-free supernatant was evaluated since it has been previously tested on *S. epidermidis* RP62A biofilm formation, demonstrating that it significantly reduced aggregation in this process; moreover, genome sequencing of this strain revealed the presence of putative biosynthetic gene clusters that might be involved in biofilm destabilization [20]. The last strain selected was *Pseudoalteromonas haloplanktis* TAD2020; the cell-free supernatant of this marine bacteria can interfere with *S. epidermidis* biofilm formation (unpublished results). The ability of TAE6080,TAD2020 and TAE2020 to produce and secrete anti-biofilm and anti-virulence molecules confirmed that Antarctic marine bacteria have great potential as a source of bioactive compounds.

However, it is by no means obvious that these bacteria could produce anti-biofilm molecules effective against ESKAPE, considering the characteristics of these pathogens. A limited number of research papers report the discovery of anti-biofilm agents effective against ESKAPE pathogens.

### 2.3. Effect of Antarctic Cell-Free Supernatants on Biofilm

The culture media used for the fermentation of Antarctic strains were synthetic, minimal, and with a defined composition. For each bacterial strain, a specific growth medium was used. The choice to use chemically defined media was to reduce the complexity of the supernatant post bacterial fermentation. For each strain, we previously set up the best medium in terms of the specific growth rate and low complexity; for the two Pseudoalteromonadales we formulated GG [27] while for the *Psychrobacter* sp. TAE2020 the most simple medium resulted to be GLUT [26], in the case of *Pseudomonas* sp. TAE6080 the best solution was GLUC [20].

The anti-biofilm effects of Antarctic bacterial culture supernatants (SNs) were examined on ESKAPE pathogens either during biofilm development by adding it to the medium at the beginning of the growth (time zero, pre-adhesion period), or after biofilm formation (24 h of bacterial culture, mature biofilm). Firstly, to exclude the effect of supernatants on bacterial viability, SNs were analyzed also for antimicrobial activity. Results did not highlight any antimicrobial activity on all ESKAPE tested strains.

As plausible, we found that biofilm growth was influenced by the saline concentration of the medium for all of the ESKAPE species (Table 7). To minimize this effect, as a control, we grew ESKAPE strains in the same medium used for the cultures of Antarctic bacteria. Then, SNs derived from the different Antarctic bacteria were diluted 1:2 in a BHI medium opportunely prepared at a concentration twice that of use.

The anti-biofilm effect was reported as the percentage of residual biofilm after treatment in comparison with untreated bacteria grown in the same culture medium (Figure 1, Figure 2, Figure 3, Figure 4, Figure 5 and Figure 6). In particular, to analyze the effect of TAE6080 SN bacteria were grown in BHI and GLUC medium; to analyze the effect of TAE2020 SN bacteria were grown in BHI and GLUT medium; and bacteria were grown in BHI and GG medium to analyze the effect of TAC125 and TAD2020 SNs. In some cases, an increase in biofilm formation was highlighted after the treatment.

#### 2.3.1. *Enterococcus*

All *Enterococcus* strains were strong biofilm producers (Table 7). The effect of each SN on biofilm formation was strongly strain-dependent (Figure 1A). SN deriving from *P. haloplanktis* TAC125 (TAC125) culture induced a reduction in biofilm production on six out of ten tested strains, with a percentage of reduction ranging between 25 and 60%. On the contrary, it notably increased the biofilm production of the *Enterococcus* EF3 strain. In previous work we reported the activity of the supernatant derived from *P. haloplanktis* TAC125 on the biofilm formation by *S. epidermidis* [25]. We identified the molecule responsible for this activity in a long-chain fatty aldehyde, a pentadecanal [18], and then we characterized and defined some derivatives of it [28]. To exclude that the antibiofilm activity of the supernatant extracted from TAC125 could be due to the pentadecanal, some experiments were performed. The activity of pentadecanal and its derivative (the pentadecanoic acid) on the biofilm formation of the *Enterococcus* strains was analyzed. Obtained results allow us to exclude any anti-biofilm activity due to these compounds.

Moreover, treatment with SN deriving from *P. haloplanktis* TAD2020 (TAD2020) induced a reduction in biofilm formation in six out of ten tested strains, although they were not the same strains influenced by TAC125. In this case, an increase was observed for *Enterococcus* AC1 and 190. Supernatants deriving from *Psychrobacter* sp. TAE2020 (TAE2020) and *Pseudomonas* sp. TAE6080 (TAE6080) caused a reduction in four out of ten strains with percentages ranging between 40 and 50%. No significant effect was observed on the mature biofilm (Figure 1B).

#### 2.3.2. *S. aureus*

*S. aureus* strains can be divided into strong, medium, and weak biofilm producers (Table 7). In Figure 2A the effect of each SN on *S. aureus* biofilm formation was reported.

The supernatant from *P. haloplanktis* TAC125 produced a reduction in biofilm formation for all tested strains except for SA12, which showed a strong increase. This strain, which did not respond to any treatment, is the weakest biofilm producer, especially in culture media with high concentrations of glutamate (GG and GLUT media). TAC125 supernatant reduced more than 85% of the biofilm on at least five strains.

Additionally, in this case, the activity of pentadecanal and pentadecanoic acid on biofilm formation of *S. aureus* strains was assessed. The obtained results allow us to exclude any anti-biofilm activity due to these compounds, except for the strain SA18 where a significant reduction in biofilm was observed (Figure 3).

A strain-dependent effect is clearly visible with supernatants derived from *P. haloplanktis* TAD2020 and *Pseudomonas* sp. TAE6080, which induced a positive response in about 50% of the tested strains (Figure 2A). *Psychrobacter* sp. TAE2020 supernatant showed a strong reduction in seven out of ten strains, while a considerable increase was observed for two strains. However, these two strains are weak biofilm producers. It is worth noting that three supernatants (TAC125, TAD2020, TAE2020) strongly impaired biofilm production in two overproducer strains, *S. aureus* ATCC6538P and *S. aureus* SA17. The activity of the supernatants on mature biofilm was extremely variable. For example, supernatants deriving from TAC125 and TAD2020 led to an increase or no effect in seven of eight clinical strains (Figure 2B).

#### 2.3.3. *K. pneumoniae*

Regarding *Klebsiella*, except for the ATCC13883 strain, all the other strains were medium or strong biofilm producers. Unexpectedly, biofilm production by the ATCC13883 strain increased about eight times in the GLUC medium, while no dramatic changes were observed for the other strains (Table 7).

The effect of supernatants on the biofilm formation of *K. pneumoniae* is reported in Figure 4A. The supernatant derived from the *Pseudomonas* sp. TAE6080 culture was very effective on *K. pneumoniae* ATCC13883, but ineffective on clinical strains (a slight effect with a 25% reduction in biofilm formation was observed only on the 1-KPC strain). The supernatant deriving from *P. haloplanktis* TAC125 impaired biofilm formation in seven out of ten strains with the percentage of reduction ranging between 55 and 70%. This anti-biofilm activity is independent of the pentadecanal and the pentadecanoic presence. The supernatant deriving from *P. haloplanktis* TAD2020 was effective on eight out of ten strains with an efficacy higher than 50% in biofilm reduction (achieving an 80% reduction in the KPC-3R strain). Moreover, the supernatant deriving from the *Psychrobacter* sp. TAE2020 culture reduced biofilm formation in seven out of ten strains, but with a lower efficacy compared to the supernatant derived from TAD2020 (the maximum activity was about 45% on 4-KPC strain). The TAE6080 supernatant was effective only on two strains, one of which was the ATCC13883 strain. Notably, the ATCC13883 strain produced a large amount of biofilm in the presence of GLUC medium, the growth medium properly used for the *Pseudomonas* sp. TAE6080 culture. All the cell-free supernatants were ineffective on mature biofilm of all *K. pneumoniae* strains, except for the ATCC1388 strain and on the pan-drug resistant KPC-3R strain (Figure 4B). The latter was one of the major biofilm producers under all tested conditions and was strongly influenced by the treatment during the early stages of biofilm formation.

#### 2.3.4. *A. baumannii*

All *A. baumannii* strains are weak biofilm producers except for the AB12 strain, which is also sensitive to all tested antibiotics (Table 7). The analyzed supernatants, in most cases, were not able to reduce biofilm formation, except in the strongest producer strain Ab12 (Figure 5A). On the other hand, it is worth noting the effect of the supernatant derived from the TAE6080 culture, which caused an increase in biofilm in all strains, sometimes even exceeding 250%. A marginal effect is found with supernatant derived from TAD2020 in seven out of ten strains with a reduction ranging from 25 to 65%.

Supernatants seemed to be active against mature biofilm, promoting its disaggregation, except for the strongest biofilm producer, the Ab12 strain (Figure 5B). The best results were obtained with supernatant derived from TAE2020 effective in seven out of ten strains, with a reduction in the mature biofilm up to 80%.

#### 2.3.5. *P. aeruginosa*

*P. aeruginosa* strains range between medium and strong biofilm producers (Table 7). Apart from a few exceptions (PA4 treated with supernatants derived from TAC125 and TAD2020, and PA16 treated with supernatants derived from TAE6080), in the other strains all supernatants reduced or, at most, left the biofilm unchanged (Figure 6). It should be noted that two supernatants, that significantly increased the biofilm produced by PA4 strain, were derived from bacterial cultures grown in GG medium, which induced a significant increase in biofilm production in the PA4 strain itself (from OD590 nm 0.816 ± 0.169 to OD590 nm 3.031 ± 0.608).

The positive results obtained by treating biofilm in the pre-adhesion stage with the supernatant derived from TAE2020 are not surprising, since in a previous work the same SN had proved to be strongly active on various *P. aeruginosa* virulence factors including biofilm formation [22]. The previous study was conducted on clinical strains isolated from cystic fibrosis patients with chronic airways infections, while strains used in this work were derived mainly from acute infections. Therefore, these results should be intended not only as a confirmation, but also as a reinforcement of the previous results, and it will be very interesting to identify the molecules responsible for these anti-virulence activities. In conclusion, supernatants deriving from TAC125 and TAD2020 bacterial cultures were very promising, not only for their action on biofilm formation but also for the treatment of mature ones. Experiments performed by using pentadecanal and pentadecanoic acid allowed us to exclude any effects due to these molecules.

#### 2.3.6. *E. coli*

As reported in Table 7, *E. coli* strains were weak producers, except for the ECNDM1 and ECNDM5 strains. The best results were obtained with supernatant derived from TAE6080, where a reduction ranging from 32 to 47%, in biofilm formation was recorded in six out of ten strains, with the best results on the strongest biofilm-producing strains (Figure 7A).

Regarding the ability to disrupt preformed biofilm, the obtained results are particularly interesting (Figure 7B). Supernatants derived from TAC125 and TAE2020 cultures were active on almost all tested strains with a breakdown of the EC1 strain up to 78% and 67% for TAC125 and TAE2020, respectively. These two supernatants were also active on the strongest biofilm producer ECNDM1. Additionally, in this case, we can exclude any effects due to pentadecanal and the pentadecanoic presence.

## 3. Discussion

Due to the resistance of ESKAPE bacteria to a broad range of antibiotics, infections sustained by these species are very difficult to eradicate, especially when they form a biofilm [29]. It is well known that biofilm-forming bacteria are about 1000 times more resistant to antimicrobials compared to planktonic cells [30]. For this reason, biofilm-associated bacteria, in particular the ESKAPE pathogens, represent a serious medical challenge worldwide. Consequently, there is an urgent need to develop new weapons to fight these pathogens, with particular emphasis on the eradication of their biofilms [31]. Several strategies are currently being explored in order to treat ESKAPE-related biofilms [2,32]. Despite this, there are no available molecules that are actually able to interfere with biofilm formation or promote biofilm disaggregation. In fact, research is mainly focused on the discovery of novel antibiotics and/or on studies of synergy with existing antibiotics so as to counter life-threatening infections.

In the search for new effective activities against pathogens, natural compounds represent more efficient products than chemically synthesized ones, with less resistance and lower side effects [31,33].

We focused our attention on cold-adapted marine bacteria deriving from Antarctica for the discovery of new anti-biofilm compounds [19]. This exotic and unusual ecological niche holds great potential as a possible source of novel drugs. Moreover, microorganisms that inhabit this environment possess a wide range of metabolic capabilities due to the physical and chemical conditions that rapidly change in this ecosystem, forcing them to abruptly adapt. For example, some molecules obtained from these bacteria display antifouling, antimicrobial, and other activities interesting for possible pharmaceutical applications [13,34,35].

The main routes for ESKAPE infections are undoubtedly medical devices such as central venous catheters, endotracheal tubes, vesical catheters, tracheostomy tubes, and percutaneous endoscopic gastrostomy, which are easily colonized by these bacteria which are ubiquitous and are members of human physiological microbiota. The prospect to chemically modify the surfaces of these devices with molecules able to impair bacterial adhesion could be a promising prevention strategy. Consequently, we are searching for molecules able to inhibit biofilm development, to prevent nosocomial infections. Furthermore, it is well-known that once a biofilm has been established, it is very difficult to eradicate. In fact, in the literature, few molecules are reported as capable of acting in the disintegration of a preformed biofilm [23,36]. For this reason, we also searched for molecules able to destabilize mature biofilms.

Biofilm quantification was assessed by using crystal violet-based assay. In the literature, various methods have been reported that quantify total biofilm or different components of biofilm [37,38].

Different methods allow the quantification of total biomass, total amount of bacterial cells, viable cell number, and amount of extracellular polymeric substances. However, these methods are often confusedly used, leading to discrepancies and misleading results.

Crystal violet staining is a reliable method for total biomass quantification. Although crystal violet binds mainly to the biofilm matrix and does not allow distinguishing between viable or dead cells, it exhibited high reproducibility and repeatability and allowed us to rapidly analyze multiple samples simultaneously [39].

In light of these reports, we think that the screening of antibiofilm activity against a large number of bacterial strains can be performed with crystal violet.

The obtained results clearly showed that any of the tested supernatants are active against all tested bacterial species. Furthermore, as expected, the activity profiles of biofilm inhibition and disaggregation are profoundly different. This result certainly confirms the heterogeneity of clinical strains, even those belonging to the same genera and species, but also suggests the different composition of supernatants deriving from the four Antarctic bacteria. The obtained results underline the deep differences between reference strains and clinical ones. This makes the need for experimental approaches such as those proposed in this work more evident, and also underlines the biofilm development complexity. Different microorganisms, although belonging to the same species, respond completely differently to treatment with the same sample. Additionally, the understanding of the mechanisms responsible for the increase in biofilm observed in such cases is certainly useful and requires a dedicated study.

Some supernatants contain molecules active both on biofilm formation and on mature biofilm disaggregation. Undoubtedly, in many cases, different molecules act on the two stages of biofilm formation; for example, supernatants deriving from TAC125 have no activity on biofilm formation by *Enterococcus* EF3 but contain at least one molecule able to disaggregate its biofilm. Similarly, the supernatant from TAE6080 is effective in preventing the biofilm formation of EF1, but it is ineffective on mature biofilm (Figure 1).

Although it could be interesting to purify the active molecules in order to gain a complete understanding of the reported results, a general overview of the reported results indicates that the anti-biofilm activity is more pronounced on Gram-negative bacterial species (Figure 8), and that for each of the 60 tested clinical ESKAPE strains, at least one supernatant effective on biofilm inhibition and/or on mature biofilm was identified, except for the mature biofilm produced by PA27 or EC8739. This very promising result confirms Antarctic marine bacteria as a valuable source of anti-biofilm molecules.

It must also be considered that infections associated with ESKAPE bacteria are often due to multispecies biofilm where different bacteria aggregate and proliferate simultaneously. For example, *K. pneumoniae*, *A. baumannii*, *P. aeruginosa*, *S. aureus*, and *Enterobacter* species are all associated with respiratory infections and pneumonia, while *K. pneumoniae*, *A. baumannii*, *P. aeruginosa*, and *Enterobacter* species are involved in urinary infections [29].

For this reason, we evaluated the activity of each supernatant on each bacterial species in order to identify samples that can simultaneously act on multiple targets in multi-species infection (Figure 9). For example, the SNs deriving from TAD2020 and TAC125 were able to disaggregate biofilms produced by *K. pneumoniae*, *A. baumannii*, and *Enterobacter* species, all involved in urinary infections.

Therefore, the possibility of using molecules capable of contrasting multi-species biofilms due to their simultaneous action on different bacterial species is certainly of interest.

## 4. Materials and Methods

### 4.1. Bacterial Strains and Growth Conditions

Different synthetic media were used for Antarctic bacteria growths. In detail, *P. haloplanktis* TAC125 and *P. haloplanktis* TAD2020 were grown in GG medium containing 10 g/L L-glutamate and 10 g/L D-gluconate as single carbon and nitrogen sources. *Psychrobacter* sp. TAE2020 was grown in GLUT medium containing 10 g/L L-glutamate as a single carbon and nitrogen source, while *Pseudomonas* sp. TAE6080 was cultivated in GLUC medium containing 10 g/L D-gluconate as a single carbon source. All media were complemented with a marine salt mix composed of 10 g/L NaCl, 1 g/L K_2_HPO_4_, 1 g/L NH_4_NO_3_, 200 mg/L MgSO_4_·7H_2_O, 5 mg/L FeSO_4_·7H_2_O, and 5 mg/L CaCl_2_·2H_2_O resulting in a final pH of 7.5 [27]. Cultures were performed in planktonic conditions at 15 °C under vigorous agitation (180 rpm) until the stationary phase of growth (72 h). The supernatant was recovered by centrifugation at 7000 rpm at 4 °C for 30 min, sterilized by filtration through membranes with a pore diameter of 0.22 μm, and stored at 4 °C until use.

The antimicrobial and antibiofilm activity of each supernatant was determined on a panel of 60 clinical and reference isolates, including *Enterococcus* species, *S. aureus*, *K. pneumoniae*, *A. baumannii*, *P. aeruginosa,* and *Enterobacter* species. The resistance profiles for each strain were performed according to the guidelines of the EUCAST Clinical Breakpoint Tables v. 13.0 (valid from 1 January 2023).

All the isolates were stored in frozen glycerol stocks, streaked on fresh Brain Heart Infusion agar plates (BHI, Oxoid, Basingstoke plate), incubated at 37 °C for 18 h, and sub-cultured under vigorous agitation (180 rpm) to provide fresh colonies; biofilm formation was performed in static conditions.

### 4.2. Biofilm Formation

#### 4.2.1. Pre-Adhesion Period

Biofilm production was quantified by adopting microtiter plate biofilm assay (MTP) as previously reported [21]. Briefly, 100 µL of overnight bacterial cultures diluted 1:100 in fresh BHI (about 0.5 OD 600 nm) were added to the wells of a sterile 96-well flat-bottomed polystyrene plate. As a control, the first row contained the untreated bacterial cells in BHI broth supplemented with the medium used for the growths of the Antarctic bacteria diluted 1:2. In detail, GG medium was used as the control for *P. haloplanktis* TAC125 and *P. haloplanktis* TAD2020 supernatant analysis; GLUC medium was used as the control for *Pseudomonas* sp. TAE6080 supernatant analysis; and GLUT medium was used as the control for *Psychrobacter* sp. TAE2020 supernatant analysis, respectively.

In the second row, the same bacterial culture was added with each supernatant at a dilution of 1:2 in BHI. The plates were aerobically incubated at 37 °C overnight. After the incubation, the contents of each well were aspirated and washed three times with distilled water to remove planktonic cells. Then, for the quantification of biofilm formation, each well was stained with 100 µL of 0.1% crystal violet and incubated for 15 min at room temperature. The excess of crystal violet was discarded, and the plates were washed three times with distilled water. The remaining dye attached to the adherent cells was solubilized with 20% (*v*/*v*) glacial acetic acid and 80% (*v*/*v*) ethanol. The total biofilm biomass in each well was spectrophotometrically quantified at 590 nm. Each data point is composed of 4 independent experiments, each performed with at least 6 replicates.

#### 4.2.2. Mature Biofilm

Assays on preformed biofilm were also performed. Briefly, 100 µL of overnight bacterial cultures, diluted 1:100 in fresh BHI (about 0.5 OD 600 nm), were added to the wells of a sterile 96-well flat-bottomed polystyrene plate. The plates were aerobically incubated for 24 h at 37 °C. Then, the content of each well was poured off and the wells were washed with sterile distilled water to remove the unattached bacteria; 100 μL of fresh medium containing each supernatant diluted 1:2 was added into each well. As controls, 100 μL of fresh BHI containing GG, GLUT, and GLUC media diluted 1:2 was separately added. The plates were again incubated for an additional 24 h (48 h in total) at 37 °C. After 24 h the plates were analyzed as previously described.

## 5. Conclusions

In conclusion, this study was focused on exploring the potential of Antarctic marine bacteria as a source of anti-biofilm molecules to fight ESKAPE pathogen infection. Four Antarctic bacterial strains were chosen, and their cell-free supernatants were tested against 60 clinical ESKAPE isolates. The supernatants did not exhibit antimicrobial activity but effectively prevented biofilm formation and dispersed mature biofilms. In detail, the obtained data showed that any of the tested supernatants is active against all ESKAPE species and that, as expected, the activity profiles of biofilm inhibition and disaggregation are profoundly different and specifically strain- and species-dependent. This result certainly confirms the heterogeneity of clinical strains but also suggests the different composition of supernatants deriving from the four Antarctic bacteria. Moreover, the evaluation of the activity of each supernatant on each pathogenic species allowed the identification of Antarctic samples that can simultaneously act on multiple targets in multi-species infections.

In light of the data shown, the message of this study is to pursue the search for anti-virulence molecules, even from unexpected sources, in order to identify new weapons against infections sustained by ESKAPE bacteria. These new activities could represent the starting point for the identification of promising new drugs that can be used in synergy with conventional antibiotics for the eradication of ESKAPE-associated infections. Given the challenges, any potential solution must be explored.

## Figures and Tables

**Figure 1 antibiotics-12-01556-f001:**
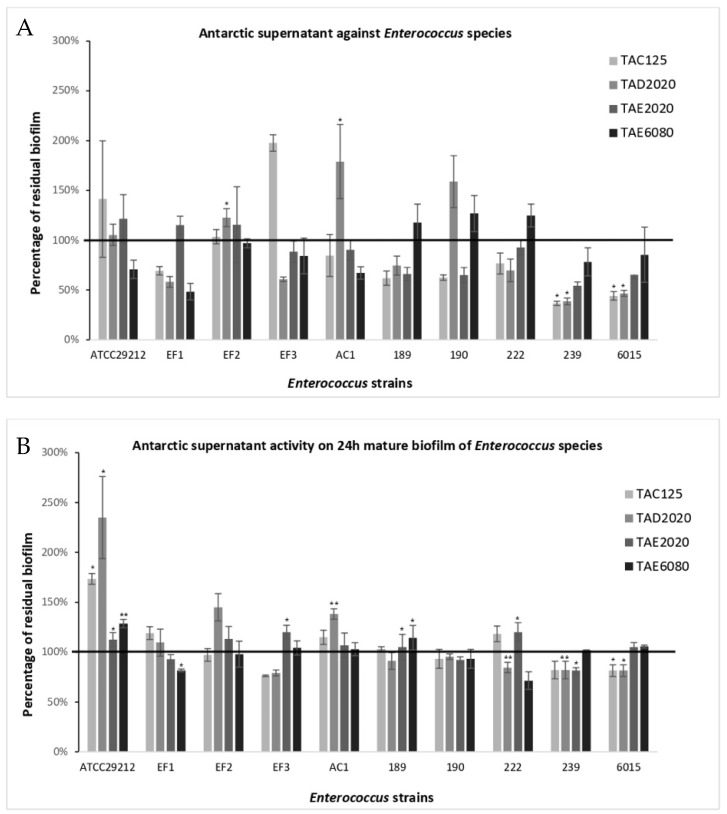
Effect of SNs from Antarctic bacteria against biofilm formation of different clinical strains and ATCC 29212 reference strain of *Enterococcus* species. Panel (**A**): Effect of SNs on biofilm formation. SNs were added to the culture medium at time zero (0 h, pre-adhesion period) and the biofilm was analyzed after overnight incubation. In the ordinate axis, the percentage of bacterial biofilm production is reported. Panel (**B**): Effect of SNs on the mature biofilm. SNs were added to the culture medium after 24 h of biofilm growth (24 h of bacterial culture) and the biofilm was analyzed after overnight incubation. In the ordinate axis, the percentage of residual biofilm is reported. Data are expressed as the percentage of residual biofilm after 24 h of treatment with SNs compared with the control sample. Each data point is composed of three independent experiments, each performed with at least three replicates. Error bars indicate the standard deviations of all the measurements. Statistical difference was determined by Student’s *t*-test: * *p* < 0.05; ** *p* < 0.01, compared with the control.

**Figure 2 antibiotics-12-01556-f002:**
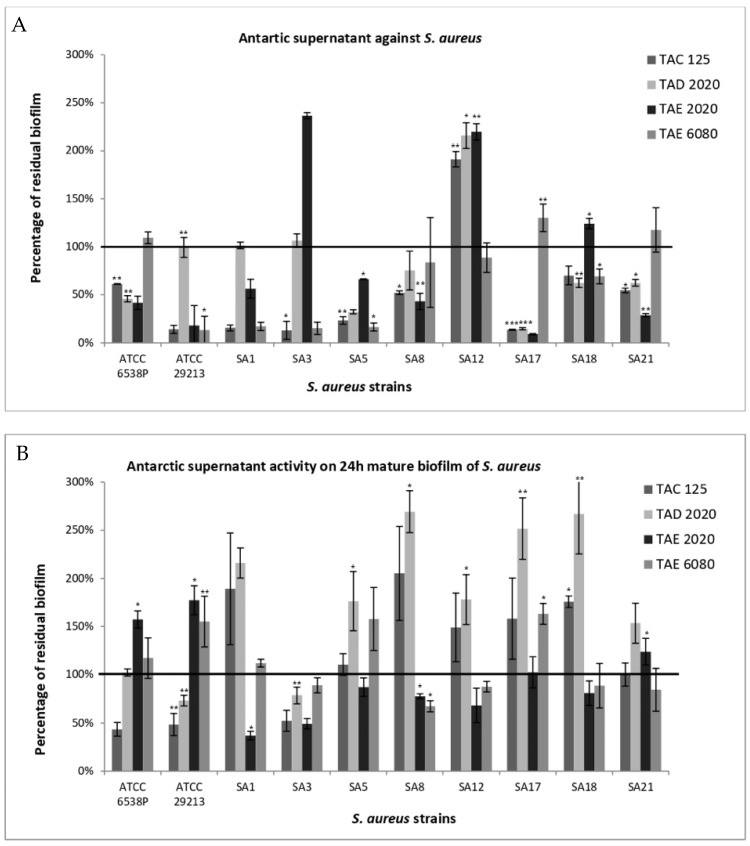
Effect of SNs from Antarctic bacteria against biofilm formation of different clinical strains and ATCC 6538P and ATCC 29213 reference strains of *S. aureus*. Panel (**A**): Effect of SNs on biofilm formation. SNs were added to the culture medium at time zero (0 h, pre-adhesion period) and the biofilm was analyzed after overnight incubation. In the ordinate axis, the percentage of bacterial biofilm production is reported. Panel (**B**): Effect of SNs on the mature biofilm. SNs were added to the culture medium after 24 h of biofilm growth (24 h of bacterial culture) and the biofilm was analyzed after overnight incubation. In the ordinate axis, the percentage of residual biofilm is reported. Data are expressed as the percentage of residual biofilm after 24 h of treatment with SNs compared with the control sample. Each data point is composed of three independent experiments, each performed with at least three replicates. Error bars indicate the standard deviations of all the measurements. Statistical difference was determined by Student’s *t*-test: * *p* < 0.05; ** *p* < 0.01, *** *p* < 0.001 compared with the control.

**Figure 3 antibiotics-12-01556-f003:**
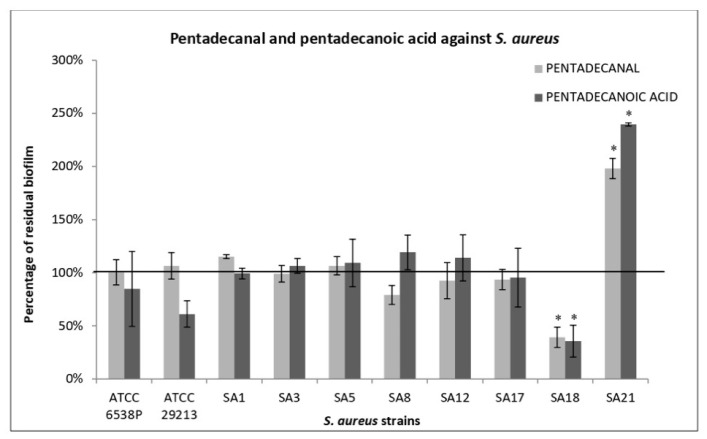
Anti-biofilm activity of pentadecanal and pentadecanoic acid on *S. aureus* biofilm formation. The anti-biofilm effect was evaluated on the biofilm formation of ATCC and clinical *S. aureus* strains using 100 µg/mL of the tested molecules. The data are reported as percentages of residual biofilm. Each data point represents the mean  ±  the SD of three independent samples. Statistical difference was determined by Student’s *t*-test: * *p* < 0.05.

**Figure 4 antibiotics-12-01556-f004:**
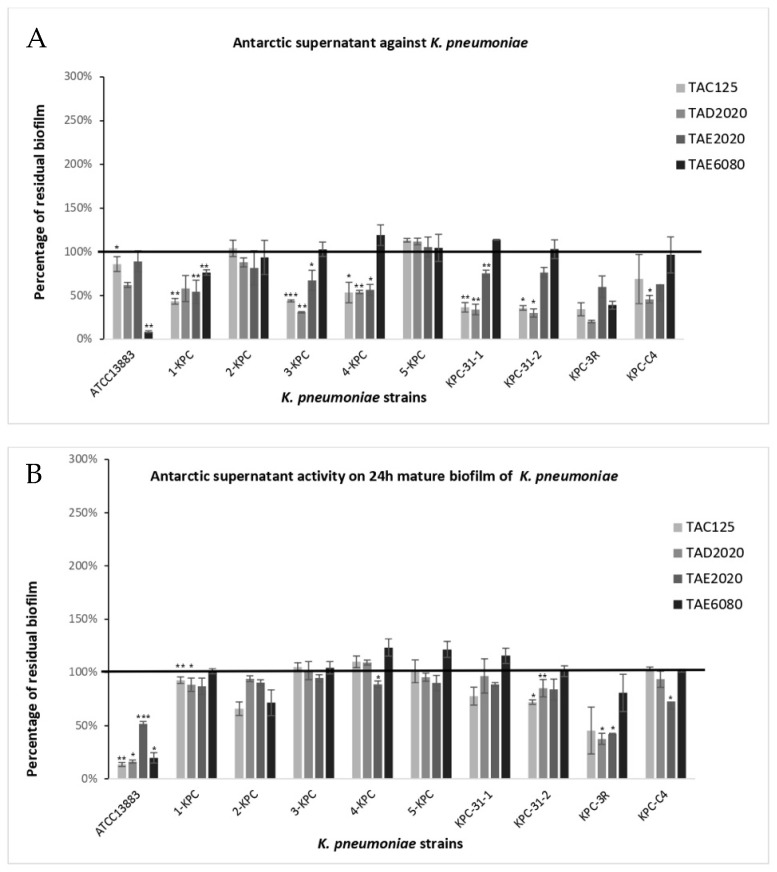
Effect of SNs from Antarctic bacteria against biofilm formation of different clinical strains and ATCC 13883 reference strain of *K. pneumoniae*. Panel (**A**): Effect of SNs on biofilm formation. SNs were added to the culture medium at time zero (0 h, pre-adhesion period) and the biofilm was analyzed after overnight incubation. In the ordinate axis, the percentage of bacterial biofilm production is reported. Panel (**B**): Effect of SNs on the mature biofilm. SNs were added to the culture medium after 24 h of biofilm growth (24 h of bacterial culture) and the biofilm was analyzed after overnight incubation. In the ordinate axis, the percentage of residual biofilm is reported. Data are expressed as the percentage of residual biofilm after 24 h of treatment with SNs compared with the control sample. Each data point is composed of three independent experiments, each performed with at least three replicates. Error bars indicate the standard deviations of all the measurements. Statistical difference was determined by Student’s *t*-test: * *p* < 0.05; ** *p* < 0.01, *** *p* < 0.001 compared with the control.

**Figure 5 antibiotics-12-01556-f005:**
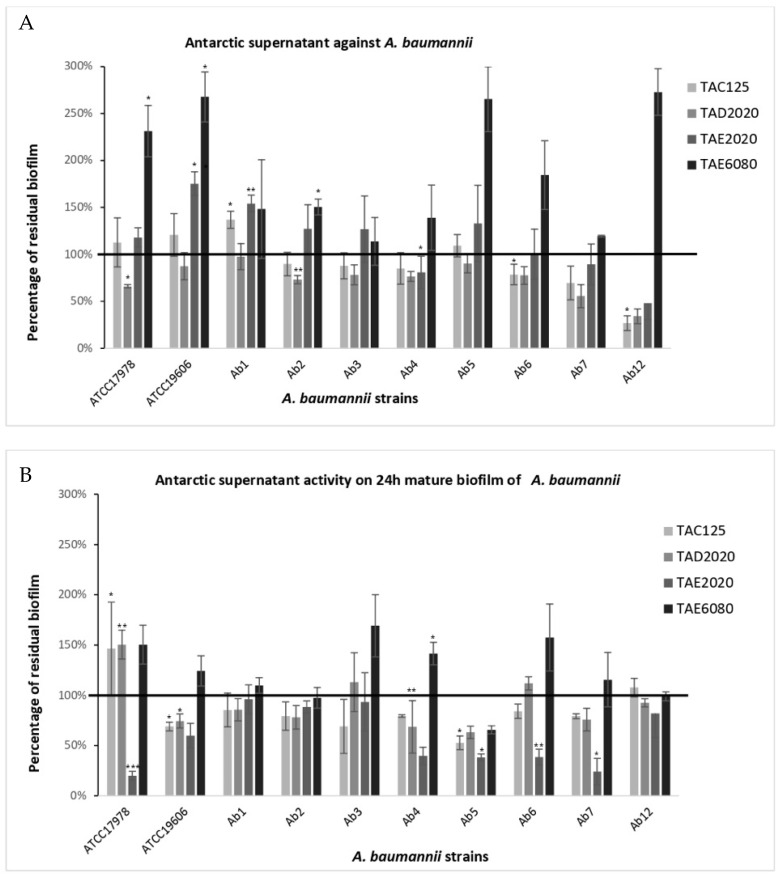
Effect of SNs from Antarctic bacteria against biofilm formation of different clinical strains and ATCC 17978 and ATCC 19606 reference strains of *A. baumannii*. Panel (**A**): Effect of SNs on biofilm formation. SNs were added to the culture medium at time zero (0 h, pre-adhesion period) and the biofilm was analyzed after overnight incubation. In the ordinate axis, the percentage of bacterial biofilm production is reported. Panel (**B**): Effect of SNs on the mature biofilm. SNs were added to the culture medium after 24 h of biofilm growth (24 h of bacterial culture) and the biofilm was analyzed after overnight incubation. In the ordinate axis, the percentage of residual biofilm is reported. Data are expressed as the percentage of residual biofilm after 24 h of treatment with SNs compared with the control sample. Each data point is composed of three independent experiments, each performed with at least three replicates. Error bars indicate the standard deviations of all the measurements. Statistical difference was determined by Student’s *t*-test: * *p* < 0.05; ** *p* < 0.01, *** *p* < 0.001 compared with the control.

**Figure 6 antibiotics-12-01556-f006:**
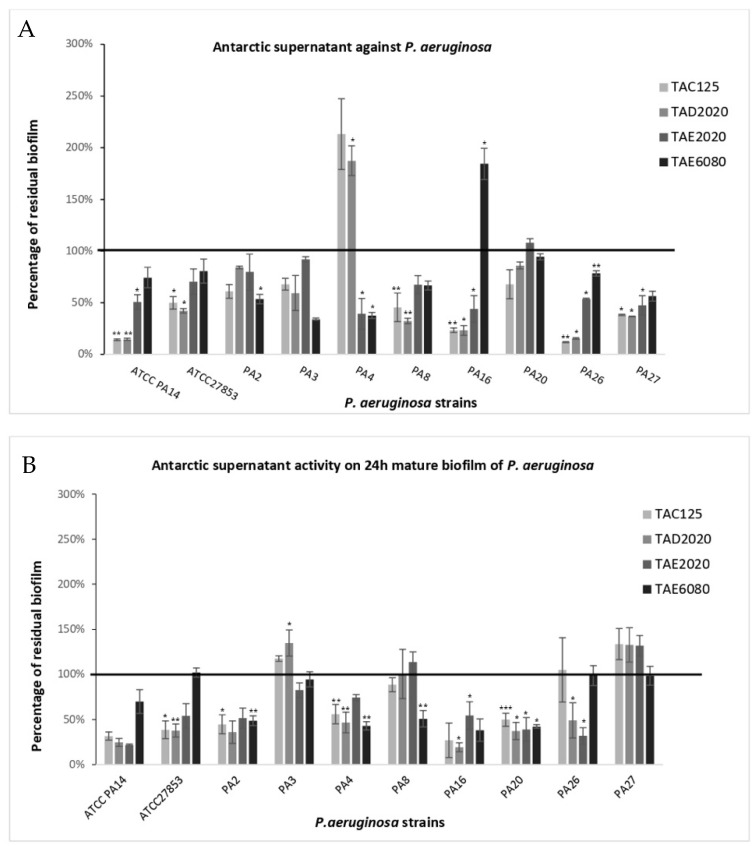
Effect of SNs from Antarctic bacteria against biofilm formation of different clinical strains and ATCC PA14 and ATCC 27853 reference strains of *P. aeruginosa*. Panel (**A**): Effect of SNs on biofilm formation. SNs were added to the culture medium at time zero (0 h, pre-adhesion period) and the biofilm was analyzed after overnight incubation. In the ordinate axis, the percentage of bacterial biofilm production is reported. Panel (**B**): Effect of SNs on the mature biofilm. SNs were added to the culture medium after 24 h of biofilm growth (24 h of bacterial culture) and the biofilm was analyzed after overnight incubation. In the ordinate axis, the percentage of residual biofilm is reported. Data are expressed as the percentage of residual biofilm after 24 h of treatment with SNs compared with the control sample. Each data point is composed of three independent experiments, each performed with at least three replicates. Error bars indicate the standard deviations of all the measurements. Statistical difference was determined by Student’s *t*-test: * *p* < 0.05; ** *p* < 0.01, *** *p* < 0.001 compared with the control.

**Figure 7 antibiotics-12-01556-f007:**
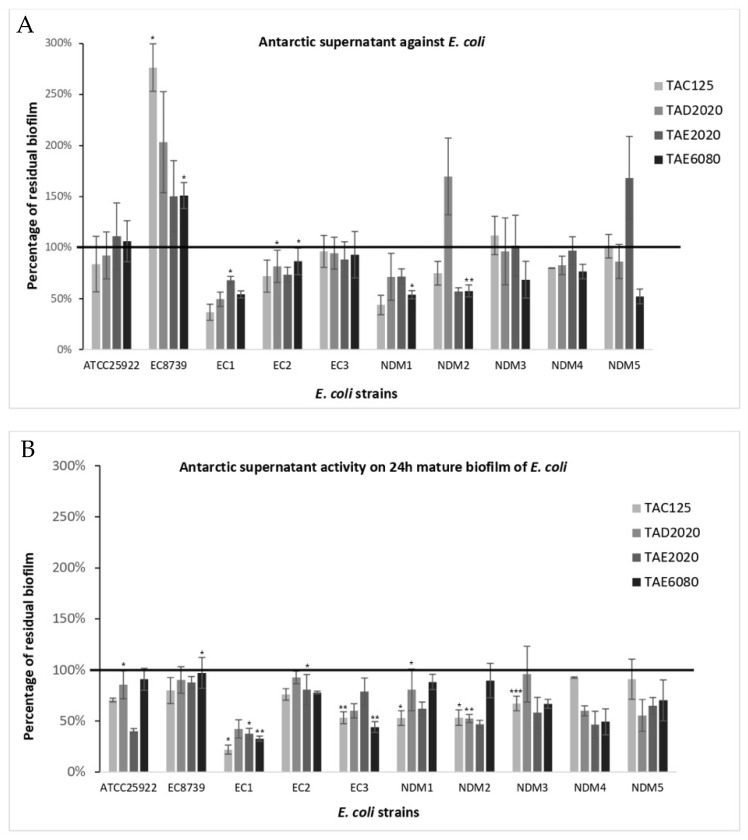
Effect of SNs from Antarctic bacteria against biofilm formation of different clinical strains and ATCC 25922 and ATCC EC8739 reference strains of *E. coli*. Panel (**A**): Effect of SNs on biofilm formation. SNs were added to the culture medium at time zero (0 h, pre-adhesion period) and the biofilm was analyzed after overnight incubation. In the ordinate axis, the percentage of bacterial biofilm production is reported. Data are expressed as the percentage of biofilm formed in the presence of SNs compared with the untreated bacteria. Each data point is composed of three independent experiments, each performed at least in three replicates. Panel (**B**): Effect of SNs on the mature biofilm. SNs were added to the culture medium after 24 h of biofilm growth (24 h of bacterial culture) and the biofilm was analyzed after overnight incubation. In the ordinate axis, the percentage of residual biofilm is reported. Data are expressed as the percentage of residual biofilm after 24 h of treatment with SNs compared with the control sample. Error bars indicate the standard deviations of all the measurements. Statistical difference was determined by Student’s *t*-test: * *p* < 0.05; ** *p* < 0.01,*** *p* < 0.001 compared with the control.

**Figure 8 antibiotics-12-01556-f008:**
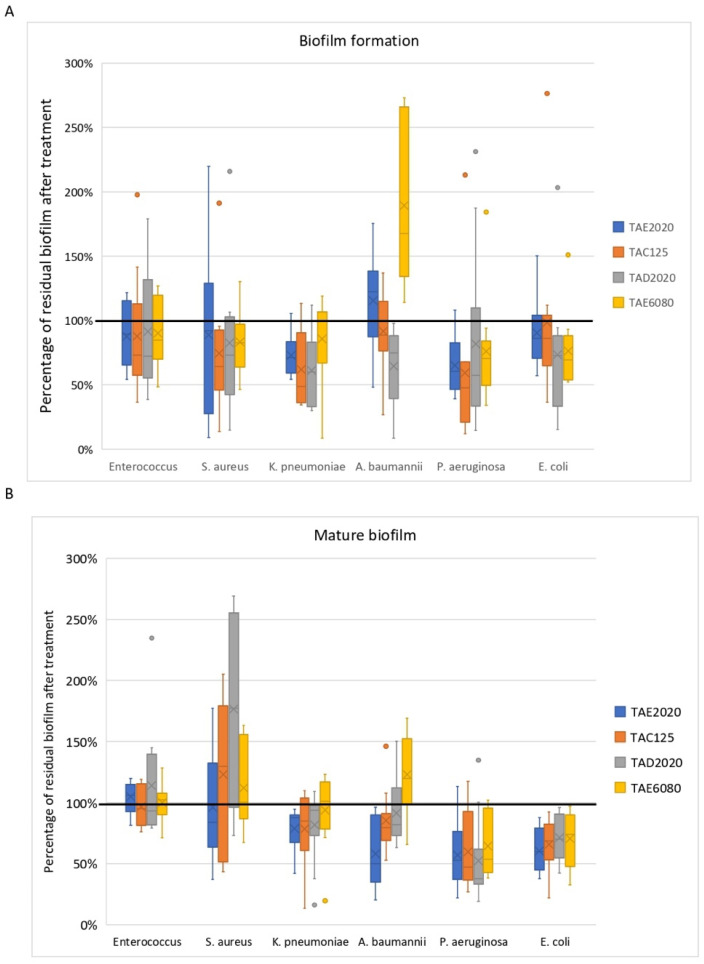
The activity of four SNs from Antarctic bacteria against each bacterial species. (**A**): Effect of SNs on biofilm formation. (**B**): Effect of SNs on the mature biofilm.

**Figure 9 antibiotics-12-01556-f009:**
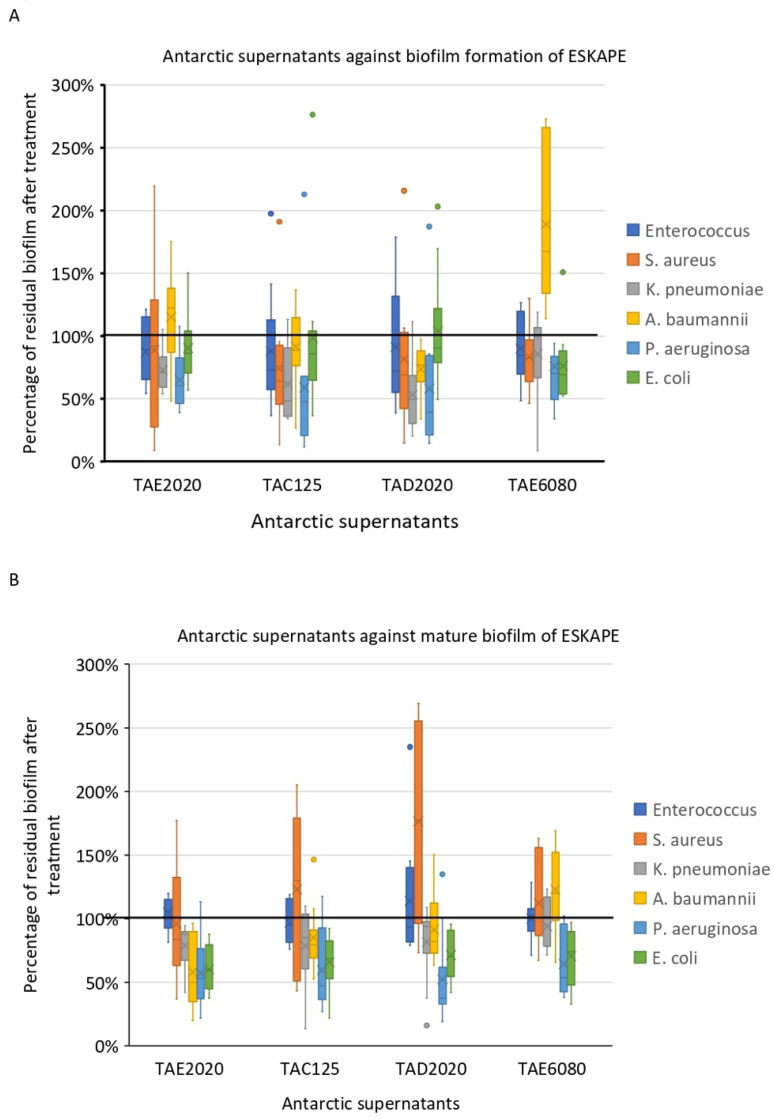
Effect of each SN from Antarctic bacteria against biofilm of all bacterial species. (**A**): Effect of SNs on biofilm formation. (**B**): Effect of SNs on the mature biofilm.

**Table 1 antibiotics-12-01556-t001:** Antimicrobial susceptibility of clinical and reference strains of *Enterococcus* species.

	Carbapenems	Fluoroquinolones		Glycopeptides		Tetracyclines	Oxazolidinones
	IM	LEV	CIP	TEC	VAN	TGC	LNZ
	(10 μg)	(5 μg)	(5 μg)	(30 μg)	(5 μg)	(15 μg)	(10 μg)
ATCC29121	I	S	S	S	S	S	R
EF1	I	S	S	S	S	S	R
EF2	I	S	S	S	S	S	S
EF3	I	S	S	S	S	S	R
AC1	I	S	S	S	S	S	R
189	I	R	R	S	S	S	R
190	I	R	R	S	S	S	S
222	I	S	S	S	S	S	R
239	I	S	S	S	S	S	R
6015	I	R	R	S	S	S	R

Antimicrobial susceptibility was performed according to the guidelines of EUCAST Clinical Breakpoint Tables v. 13.0 (valid from 1 January 2023). IM: imipenem; LEV: levofloxacin; CIP: ciprofloxacin; TEC: teicoplanin; VAN: vancomycin; TGC: tigecycline; LNZ: linezolid. S: sensitive; R: resistance; I: intermediate.

**Table 2 antibiotics-12-01556-t002:** Antimicrobial susceptibility of clinical and reference strains of *S. aureus*.

	Penicillins	Cephalosporins	Fluoroquinolones		Aminoglycosides	Tetracyclines	Glycopeptides	Miscellaneous
	OXA	FOX	LEV	CIP	AK	TET	VAN	SXT
	(2 μg/mL)	(30 μg)	(5 μg)	(5 μg)	(30 μg)	(30 μg)	(2 μg/mL)	(25 μg)
ATCC 6538P	S	S	S	S	S	S	S	S
ATCC 29213	S	S	R	R	S	S	R	S
SA1	R	S	R	-	S	S	R	S
SA3	R	S	R	-	S	S	R	S
SA5	S	S	R	-	S	S	R	S
SA8	R	R	R	R	S	S	R	S
SA12	R	R	R	R	S	S	R	S
SA17	R	R	R	R	S	S	R	S
SA18	R	R	R	-	S	S	R	S
SA21	R	R	R	R	S	S	R	S

Antimicrobial susceptibility was performed according to the guidelines of EUCAST Clinical Breakpoint Tables v. 13.0 (valid from 1 January 2023). OXA: oxacillin; FOX: cefoxitin; LEV: levofloxacin; CIP: ciprofloxacin; AK: amikacin; TET: tetracycline; VAN: vancomycin; SXT: trimethoprim/sulfamethoxazole. S: sensitive; R: resistance; -: not tested.

**Table 3 antibiotics-12-01556-t003:** Antimicrobial susceptibility of clinical and reference strains of *K. pneumoniae*.

	Penicillins		Cephalosporins		Carbapenems	Monobactams	Fluoroquinolones	Aminoglycosides	Miscellaneous
	AMP	AMC	FOX	CRO	IM	ATM	CIP	AK	SXT
	(10 μg)	(30 μg)	(30 μg)	(30 μg)	(10 μg)	(30 μg)	(5 μg)	(30 μg)	(25 μg)
ATCC13883	R	S	S	S	S	S	R	S	S
1-KPC	R	R	R	R	S	R	R	S	R
2-KPC	R	R	R	R	R	R	R	R	S
3-KPC	R	R	R	R	R	R	R	R	S
4-KPC	R	R	R	R	R	R	R	R	S
5-KPC	R	R	R	R	R	R	R	R	R
KPC31-1	R	R	S	R	S	R	R	S	R
KPC31-2	R	R	S	R	S	R	R	S	R
KPC-3R	R	R	R	R	R	R	R	R	R
KPC-C4	R	S	S	R	S	R	R	S	R

Antimicrobial susceptibility was performed according to the guidelines of EUCAST Clinical Breakpoint Tables v. 13.0 (valid from 1 January 2023). AMP: ampicillin; AMC: amoxicillin and clavulanic acid; FOX: cefoxitin; CRO: ceftriaxone; IM: imipenem; ATM: aztreonam; CIP: ciprofloxacin; AK: amikacin; SXT: trimethoprim/sulfamethoxazole. S: sensitive; R: resistance.

**Table 4 antibiotics-12-01556-t004:** Antimicrobial susceptibility of clinical and reference strains of *A. baumannii*.

	Carbapenems		Fluoroquinolones		Aminoglycosides			Miscellaneous
	MRP	IM	CIP	LEV	AK	CN	TOB	SXT
	(10 μg)	(10 μg)	(5 μg)	(5 μg)	(30 μg)	(10 μg)	(10 μg)	(25 μg)
ATCC 17978	S	S	I	S	S	S	S	I
ATCC 19606	S	S	I	S	R	R	R	R
Ab1	R	R	R	R	S	R	R	R
Ab2	R	R	R	R	S	R	R	R
Ab3	R	R	R	R	R	R	R	R
Ab4	R	R	R	R	R	R	R	R
Ab5	R	R	R	R	R	R	R	R
Ab6	R	R	R	R	R	R	R	R
Ab7	R	R	R	R	R	R	R	R
Ab12	S	S	S	S	S	S	S	S

Antimicrobial susceptibility was performed according to the guidelines of EUCAST Clinical Breakpoint Tables v. 13.0 (valid from 1 January 2023). MRP: meropenem; IM: imipenem; CIP: ciprofloxacin; LEV: levofloxacin; AK: amikacin; CN: gentamicin; TOB: tobramycin; SXT: trimethoprim/sulfamethoxazole; S: sensitive; R: resistance; I: intermediate.

**Table 5 antibiotics-12-01556-t005:** Antimicrobial susceptibility of clinical and reference strains of *P. aeruginosa*.

	Penicillins	Cephalosporins	Carbapenems		Fluoroquinolones	Aminoglycosides
	TZP	CZA	MRP	IM	LEV	AK
	(36 μg)	(14 μg)	(10 μg)	(10 μg)	(5 μg)	(30 μg)
ATCC PA14	S	S	-	S	S	S
ATCC 27853	I	S	S	R	I	S
PA2	I	S	-	I	I	S
PA3	I	S	-	I	I	S
PA4	I	S	-	I	I	S
PA8	I	S	-	I	I	S
PA16	I	S	-	I	I	S
PA20	I	S	-	I	I	S
PA26	I	S	S	R	R	S
PA27	I	S	-	I	I	S

Antimicrobial susceptibility was performed according to the guidelines of EUCAST Clinical Breakpoint Tables v. 13.0 (valid from 1 January 2023). TZP: Piperacillin-tazobactam; CZA: Ceftazidime/avibactam; MRP: meropenem; IM: imipenem; LEV: levofloxacin; AK: amikacin; S: sensitive; R: resistance; I: intermediate; -: not tested.

**Table 6 antibiotics-12-01556-t006:** Antimicrobial susceptibility of clinical and reference strains of *E. coli*.

	Penicillins		Cephalosporins		Carbapenems	Monobactams	Fluoroquinolones	Aminoglycosides	Miscellaneous
	AMP	AMC	FOX	CRO	IM	ATM	CIP	AK	SXT
	(10 μg)	(30 μg)	(30 μg)	(30 μg)	(10 μg)	(30 μg)	(5 μg)	(30 μg	(25 μg)
ATCC 25922	S	S	S	S	S	S	S	S	S
ATCC EC8739	S	S	S	S	S	S	S	S	S
EC1	R	S	S	S	S	S	S	R	R
EC2	S	S	S	S	S	S	S	S	S
EC3	S	S	S	S	S	S	R	S	R
ECNDM1	R	R	R	R	S	R	R	S	S
ECNDM2	R	R	R	R	R	R	R	S	R
ECNDM3	R	R	R	R	S	R	R	S	S
ECNDM4	R	R	R	R	R	R	R	S	R
ECNDM5	R	R	R	R	R	R	R	S	R

Antimicrobial susceptibility was performed according to the guidelines of EUCAST Clinical Breakpoint Tables v. 13.0 (valid from 1 January 2023). AMP: ampicillin; AMC: amoxicillin and clavulanic acid; FOX: cefoxitin; CRO: ceftriaxone; IM: imipenem; ATM: aztreonam; CIP: ciprofloxacin; AK: amikacin; SXT: trimethoprim/sulfamethoxazole. S: sensitive; R: resistance.

**Table 7 antibiotics-12-01556-t007:** Biofilm amount measured at 590 nm after crystal violet staining in the presence of different bacterial media.

Bacterial Species	MEDIA			
	BHI	GLUC	GLUT	GG
*Enterococcus*				
ATCC29212	0.560 ± 0.045	0.580 ± 0.125	0.414 ± 0.022	0.387 ± 0.108
EF1	0.636 ± 0.136	0.692 ± 0.207	0.314 ± 0.018	0.587 ± 0.171
EF2	0.524 ± 0.095	0.392 ± 0.065	0.369 ± 0.043	0.356 ± 0.051
EF3	0.484 ± 0.072	0.344 ± 0.038	0.232 ± 0.024	0.311 ± 0.060
AC1	0.528 ± 0.140	0.328 ± 0.093	0.293 ± 0.046	0.375 ± 0.016
189	1.581 ± 0.161	0.856 ± 0.150	0.772 ± 0.198	0.671 ± 0.139
190	1.489 ± 0.068	0.785 ± 0.185	0.665 ± 0.117	0.669 ± 0.126
222	0.638 ± 0.043	0.402 ± 0.018	0.426 ± 0.006	0.499 ± 0.064
239	1.157 ± 0.100	0.846 ± 0.191	0.523 ± 0.116	0.658 ± 0.154
6015	1.764 ± 0.142	1.105 ± 0.175	0.713 ± 0.207	1.045 ± 0.171
*S. aureus*				
ATCC 6538P	2.215 ± 0.133	1.239 ± 0.128	0.831 ± 0.093	1.063 ± 0.065
ATCC 29213	1.392 ± 0.128	0.814 ± 0.140	0.346 ± 0.062	0.293 ± 0.013
SA1	0.237 ± 0.031	0.152 ± 0.011	0.149 ± 0.008	0.163 ± 0.005
SA3	0.249 ± 0.017	0.160 ± 0.028	0.160 ± 0.018	0.154 ± 0.011
SA5	0.534 ± 0.153	0.391 ± 0.033	0.635 ± 0.133	0.582 ± 0.044
SA8	0.254 ± 0.035	0.493 ± 0.021	0.532 ± 0.052	0.368 ± 0.045
SA12	0.237 ± 0.006	0.144 ± 0.004	0.069 ± 0.0030	0.071 ± 0.009
SA17	3.230 ± 0.152	1.257 ± 0.078	2.060 ± 0.311	1.132 ± 0.042
SA18	0.332 ± 0.044	0.261 ± 0.017	0.257 ± 0.015	0.298 ± 0.064
SA21	0.495 ± 0.080	0.571 ± 0.072	0.681 ± 0.018	0.330 ± 0.038
*K. pneumoniae*				
ATCC13883	0.347 ± 0.029	2.584 ± 0.858	0.394 ± 0.077	0.396 ± 0.117
1-KPC	1.191 ± 0.067	0.869 ± 0.074	1.147 ± 0.097	1.146 ± 0.107
2-KPC	0.625 ± 0.094	0.422 ± 0.073	0.495 ± 0.044	0.498 ± 0.073
3-KPC	1.347 ± 0.073	1.056 ± 0.059	1.413 ± 0.154	1.432 ± 0.092
4-KPC	0.679 ± 0.080	0.414 ± 0.057	0.666 ± 0.044	0.701 ± 0.068
5-KPC	0.499 ± 0.062	0.375 ± 0.056	0.450 ± 0.036	0.450 ± 0.036
KPC-31-1	1.586 ± 0.162	0.850 ± 0.061	1.001 ± 0.050	0.819 ± 0.024
KPC-31-2	1.729 ± 0.146	1.175 ± 0.094	1.182 ± 0.138	1.010 ± 0.190
KPC-3R	0.531 ± 0.271	0.836 ± 0.320	0.759 ± 0.630	1.004 ± 0.490
KPC-C4	1.521 ± 0.143	0.883 ± 0.117	1.119 ± 0.143	0.964 ± 0.163
*Follows*				
*A. baumannii*				
ATCC17978	0.273 ± 0.063	0.276 ± 0.021	0.276 ± 0.083	0.298 ± 0.037
ATCC19606	0.270 ± 0.026	0.285 ± 0.042	0.217 ± 0.028	0.328 ± 0.042
Ab1	0.344 ± 0.028	0.212 ± 0.019	0.193 ± 0.009	0.212 ± 0.009
Ab2	0.295 ± 0.082	0.244 ± 0.041	0.257 ± 0.007	0.319 ± 0.027
Ab3	0.526 ± 0.066	0.258 ± 0.013	0.216 ± 0.012	0.260 ± 0.028
Ab4	0.372 ± 0.098	0.245 ± 0.005	0.268 ± 0.058	0.265 ± 0.033
Ab5	0.231 ± 0.054	0.171 ± 0.003	0.211 ± 0.020	0.224 ± 0.012
Ab6	0.210 ± 0.056	0.218 ± 0.019	0.222 ± 0.027	0.245 ± 0.017
Ab7	0.238 ± 0.023	0.256 ± 0.010	0.420 ± 0.031	0.465 ± 0.041
Ab12	0.584 ± 0.026	0.834 ± 0.317	1.691 ± 0.163	1.804 ± 0.155
*P. aeruginosa*				
ATCC PA14	0.750 ± 0.108	1.129 ± 0.031	0.953 ± 0.026	0.948 ± 0.111
ATCC27853	0.233 ± 0.021	1.093 ± 0.110	0.553 ± 0.101	0.564 ± 0.062
PA2	0.613 ± 0.119	0.612 ± 0.049	0.446 ± 0.023	0.454 ± 0.086
PA3	0.943 ± 0.222	1.993 ± 0.833	0.751 ± 0.127	1.097 ± 0.277
PA4	0.816 ± 0.169	3.031 ± 0.608	1.347 ± 0.081	1.310 ± 0.199
PA8	0.705 ± 0.201	1.664 ± 0.403	1.496 ± 0.403	1.294 ± 0.124
PA16	0.608 ± 0.059	0.419 ± 0.101	0.461 ± 0.037	0.517 ± 0.083
PA20	0.670 ± 0.161	0.512 ± 0.032	0.409 ± 0.023	0.415 ± 0.037
PA26	0.658 ± 0.107	1.113 ± 0.050	1.217 ± 0.131	1.193 ± 0.194
PA27	2.083 ± 0.338	2.347 ± 0.652	2.865 ± 0.387	2.436 ± 0.426
*E. coli*				
ATCC25922	0.332 ± 0.063	0.269 ± 0.042	0.287 ± 0.067	0.300 ± 0.067
EC8739	0.230 ± 0.025	0.179 ± 0.006	0.247 ± 0.029	0.181 ± 0.005
EC1	0.267 ± 0.079	0.435 ± 0.145	0.531 ± 0.103	0.547 ± 0.180
EC2	0.136 ± 0.020	0.167 ± 0.030	0.133 ± 0.013	0.160 ± 0.027
EC3	0.132 ± 0.024	0.132 ± 0.015	0.124 ± 0.028	0.131 ± 0.009
NDM1	0.529 ± 0.236	1.049 ± 0.287	0.841 ± 0.335	1.155 ± 0.640
NDM2	0.351 ± 0.166	0.325 ± 0.099	0.357 ± 0.146	0.277 ± 0.176
NDM3	0.175 ± 0.083	0.282 ± 0.211	0.186 ± 0.124	0.231 ± 0.132
NDM4	0.177 ± 0.087	0.255 ± 0.101	0.198 ± 0.064	0.274 ± 0.097
NDM5	0.370 ± 0.218	0.722 ± 0.270	0.420 ± 0.150	0.861 ± 0.279

## Data Availability

Data could be obtained from the corresponding author according to the request.

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
