# Peer review of "Antarctic Marine Bacteria as a Source of Anti-Biofilm Molecules to Combat ESKAPE Pathogens"

_antibiotics, 2023, doi:10.3390/antibiotics12101556_

Round 1
Reviewer 1 Report
In this study, Artini et al. reported that the supernatants from Antarctic marine bacteria could affect the biofilm formation. However, it is still a primary study in the current state.
Major comments:
1, It is strongly recommended that the author analyze the composition of the fermentation broth.
2, Does the supernatant of fermentation broth have a synergistic effect with commonly used antibiotics in inhibiting the growth of pathogenic bacteria? It is worth the author trying it out.
The authors should integrate and refine the description of the results.
Author Response
Reviewer 1
Comments and Suggestions for Authors
In this study, Artini et al. reported that the supernatants from Antarctic marine bacteria could affect the biofilm formation. However, it is still a primary study in the current state.
Major comments:
1, It is strongly recommended that the author analyze the composition of the fermentation broth.
We thank the reviewer for the comments. The culture media used for the fermentation of Antarctic strains were synthetic, minimal and with a defined composition. For each bacterial strain, a specific growth medium was used. In detail, Pseudoalteromonas haloplanktis TAC125 and Pseudoalteromonas haloplanktis TAD2020 were grown in GG medium containing 10 g/l L-glutamate and 10 g/l D-gluconate as single carbon and nitrogen sources. Psychrobacter sp. TAE2020 was grown in GLUT medium containing 10 g/l L-glutamate as a single carbon and nitrogen source, while Pseudomonas sp. TAE6080 was cultivated in GLUC medium containing 10 g/l D-gluconate as a single carbon source. All media were complemented with a marine salt mix composed of 10 g/l NaCl, 1 g/l K2HPO4, 1 g/l NH4NO3, 200 mg/l MgSO4·7H2O, 5 mg/l FeSO4·7H2O, and 5 mg/l CaCl2·2H2O resulting in a final of pH 7.5.
The choice to use chemically defined media was to reduce the complexity of supernatant post-bacterial fermentation. This approach could make easier the complex process of the identification of the molecules produced and secreted by the microbial cells and therefore the discovery of the bioactive compound.
2, Does the supernatant of fermentation broth have a synergistic effect with commonly used antibiotics in inhibiting the growth of pathogenic bacteria? It is worth the author trying it out.
We thank the reviewer for the suggestion.
However, bacterial supernatants are very complex samples containing proteins, peptides, lipids, carbohydrates, polysaccharides, and different small organic molecules. For this reason, the evaluation of synergistic effects with commonly used antibiotics could be very difficult to obtain.
Our future goal will be the identification of molecule(s) responsible for the antibiofilm activity and to test them in synergy with conventional antibiotics against pathogenic bacteria.
Comments on the Quality of English Language
The authors should integrate and refine the description of the results.
We thank the reviewer for the suggestion. Description of results has been refined and integrated as suggested.
Reviewer 2 Report
This study underscores the pressing global health threat posed by ESKAPE pathogens, renowned for their resistance to antimicrobial drugs and their role in hospital-acquired infections, particularly in intensive care units. Its focused objective of investigating the potential of Antarctic marine bacteria as a source of anti-biofilm molecules is well-articulated. Four Antarctic bacterial strains were systematically tested against a significant number of clinical ESKAPE isolates, revealing that while the bacterial supernatants lacked direct antimicrobial activity, they demonstrated efficacy in preventing biofilm formation and dispersing mature biofilms. These findings hold promise for the development of novel strategies to counter biofilms formed by clinically significant bacterial species, thereby contributing to the vital endeavor of preventing and controlling infections caused by ESKAPE pathogens. However, my only concern is with the conclusion section. Revise conclusion section.
Author Response
Reviewer 2
Comments and Suggestions for Authors
This study underscores the pressing global health threat posed by ESKAPE pathogens, renowned for their resistance to antimicrobial drugs and their role in hospital-acquired infections, particularly in intensive care units. Its focused objective of investigating the potential of Antarctic marine bacteria as a source of anti-biofilm molecules is well-articulated. Four Antarctic bacterial strains were systematically tested against a significant number of clinical ESKAPE isolates, revealing that while the bacterial supernatants lacked direct antimicrobial activity, they demonstrated efficacy in preventing biofilm formation and dispersing mature biofilms. These findings hold promise for the development of novel strategies to counter biofilms formed by clinically significant bacterial species, thereby contributing to the vital endeavor of preventing and controlling infections caused by ESKAPE pathogens. However, my only concern is with the conclusion section. Revise conclusion section.
We thank the reviewer for the suggestion. The conclusion section has been emended as reported below (lanes 813-830):
In conclusion, this study was focused on exploring the potential of marine Antarctic bacteria as a source of anti-biofilm molecules to fight ESKAPE pathogens infection. Four Antarctic bacterial strains were chosen, and their cell-free supernatants were tested against 60 clinical ESKAPE isolates. The supernatants did not exhibit antimicrobial activity but effectively prevented biofilm formation and dispersed mature biofilms. In detail, the obtained data showed that any of the tested supernatants is active against all ESKAPE species and that, as expected, the activity profiles of biofilm inhibition and disaggregation are profoundly different and specifically strain- and species- dependent. This result certainly confirms the heterogeneity of clinical strains but also suggests the different composition of supernatants deriving from four Antarctic bacteria. Moreover, the evaluation of the activity of each supernatant on each pathogenic species allowed the identification of Antarctic samples that can simultaneously act on multiple targets in multi-species infections.
In light of the data shown, the message of this study is to pursue the search for anti-virulence molecules, even from unexpected sources, in order to identify new weapons against infections sustained by ESKAPE bacteria. These new activities could represent the starting point for the identification of promising new drugs to be used in synergy with conventional antibiotics for the eradication of ESKAPE-associated infections. Given the challenge, any potential solution must be explored.
Reviewer 3 Report
The authors provide experimental data in order to demonstrate the anti-biofilm effect of four bacterial strains of Antarctic origin. The idea is possibly original since the exploration of bacteria from exotic places can help the discovery of new agents. However, the work suffers from excessive simplicity, the methods are not very decisive and the conclusions are too general.
In my opinion, some method to quantify the biofilm beyond crystal violet, as well as data on the nanomechanics of the treated biofilm compared to the untreated one, are required before publishing this paper in Antibiotics.
Author Response
Reviewer 3
Comments and Suggestions for Authors
The authors provide experimental data in order to demonstrate the anti-biofilm effect of four bacterial strains of Antarctic origin. The idea is possibly original since the exploration of bacteria from exotic places can help the discovery of new agents. However, the work suffers from excessive simplicity, the methods are not very decisive and the conclusions are too general.
In my opinion, some method to quantify the biofilm beyond crystal violet, as well as data on the nanomechanics of the treated biofilm compared to the untreated one, are required before publishing this paper in Antibiotics.
We thank the reviewer for his/her comments.
In this paper, the cell-free supernatants deriving from four Antarctic bacterial strains were tested against 60 clinical ESKAPE isolates.
In literature, various methods have been reported to quantify total biofilm or different components of biofilm.
Wilson and coworkers (2017), report:
“the crystal violet microtiter plate assay consists of several steps, it is relatively easy to perform, reproducible, and allows researchers to rapidly analyze multiple samples simultaneously. It is relatively inexpensive as it does not require the purchase of specialized equipment, and the dye is inexpensive with a shelf life of years if protected from contamination. Furthermore, the crystal violet assay can be modified for biofilms grown in a variety of reactors”.
(Wilson C, Lukowicz R, Merchant S, Valquier-Flynn H, Caballero J, Sandoval J, Okuom M, Huber C, Brooks TD, Wilson E, Clement B, Wentworth CD, Holmes AE. Quantitative and Qualitative Assessment Methods for Biofilm Growth: A Mini-review. Res Rev J Eng Technol. 2017).
Again, Stiefel and coworkers (2016) have systematically compared different methods for quantification of biofilm in microtiter plates, including those for total biomass, total amount of bacterial cells, viable cell number, and amount of extracellular polymeric substances. However, these methods are often confusedly used, leading to discrepancies and misleading results. They found that most of the methods tested in general exhibited high reproducibility and repeatability and they concluded that crystal violet staining was a simple but reliable method for total biomass quantification.
(Stiefel, P., Rosenberg, U., Schneider, J., Mauerhofer, S., Maniura-Weber, K., & Ren, Q. (2016). Is biofilm removal properly assessed? Comparison of different quantification methods in a 96-well plate system. Applied microbiology and biotechnology, 100(9), 4135–4145.)
In light of these reports, we think that the screening of antibiofilm activity against a large number of bacterial strains can be performed with crystal violet.
In light of these reports, we believe that crystal violet assay is the best solution to do a screening of antibiofilm activity against a large number of bacterial strains.
Furthermore, bacterial supernatants are very complex samples containing proteins, peptides, lipids, carbohydrates, polysaccharides, and different small organic molecules. Data on the nanomechanics of the treated biofilm compared to the untreated one could be very difficult to obtain in this condition and could lead to an incorrect interpretation of the results.
Our future goal will be the identification of molecule(s) responsible for the antibiofilm activity and then clarify the possible mechanisms of action.
Reviewer 4 Report
Overall, the authors describe the fermentation and bioactivities of the cell-free supernatants of four Antarctic bacterial isolates against clinical ESKAPE. Overall, the data are very well presented and characterized, however there are some minor revision/edits which need to be revised prior to publication as summarized below:
Comments/edits to authors:
· The authors should provide more details regarding the four selected strains (the isolation and identification of the four selected Antarctic bacterial isolates), even if this has been reported previously, they should include a paragraph in the experimental section. No background information/or details regarding the four selected bacterial isolates, and why the authors have selected only these four strains? Also – in the introduction part, the authors should highlight how they selected the four Antarctic marine bacteria to analyze the effect of supernatants derived from them? How many were the total isolated strains and how did they prioritize them to only 4 selected strains?
· The authors used three different growths media to grow the four selected Antarctic bacteria – why? And why the authors selected these three media/conditions. Did the authors check many others different media/optimization conditions for the selection of the most active cell-free supernatant for each bacterial isolate? This should be discussed/highlighted in the manuscript.
· All the activity figures are missing the controls. The culture media should be included in all the bioactivity figures as negative control. Did the authors test the culture media without any growing bacteria? The growing media could have some biological effects.
Author Response
Reviewer 4
Comments and Suggestions for Authors
Overall, the authors describe the fermentation and bioactivities of the cell-free supernatants of four Antarctic bacterial isolates against clinical ESKAPE. Overall, the data are very well presented and characterized, however there are some minor revision/edits which need to be revised prior to publication as summarized below:
Comments/edits to authors:
The authors should provide more details regarding the four selected strains (the isolation and identification of the four selected Antarctic bacterial isolates), even if this has been reported previously, they should include a paragraph in the experimental section. No background information/or details regarding the four selected bacterial isolates, and why the authors have selected only these four strains? Also – in the introduction part, the authors should highlight how they selected the four Antarctic marine bacteria to analyze the effect of supernatants derived from them? How many were the total isolated strains and how did they prioritize them to only 4 selected strains?
We thank the reviewer for the comments and the manuscript has been emended as reported below.
A paragraph in the Results section with a detailed description of Antarctic strains was added. Information on the isolation and identification of Antarctic marine bacteria was included with the corresponding references. Furthermore, the criteria of selection for these strains were highlighted. As reported in the new version of the manuscript, we chose to further investigate bacterial strains possessing previously reported antiadhesive capabilities. Moreover, the priority was also defined for bacterial strains with sequenced genomes (TAD2020 not yet published).
Below you can find the corresponding paragraph (lanes 180-199):
2.2 Selection of Antarctic Marine Bacteria
In this work, we analyzed the effect of supernatants derived from selected Antarctic marine bacteria against ESKAPE pathogens. In detail, we selected Pseudoalteromonas haloplanktis TAC125 [24] which produces a secreted molecule, the pentadecanal, able to inhibit the biofilm growth of Staphylococcus epidermidis [18,25]. The second marine strain chosen was Psychrobacter sp. TAE2020 [26]; it produces and secretes molecules active against some virulence factors of P. aeruginosa isolated from patients affected by cystic fibrosis, such as biofilm formation and accumulation, pyocyanin production and swimming and swarming motility [22]. Pseudomonas sp. TAE6080 cell-free supernatant was evaluated since it has been previously tested on S. epidermidis RP62A biofilm formation, demonstrating that it significantly reduced aggregation this process; moreover, genome sequencing of this strain revealed the presence of putative biosynthetic gene clusters that might be involved in biofilm destabilization [20]. The last strain selected was Pseudoalteromonas haloplanktis TAD2020: the cell-free supernatant of this marine bacteria can interfere with S. epidermidis biofilm formation (unpublished results). The ability of TAE6080,TAD2020 and TAE2020 to produce and secrete anti-biofilm and anti-virulence molecules confirmed that Antarctic marine bacteria have great potential as a source of bioactive compounds.
However, it is by no means obvious that these bacteria could produce anti-biofilm molecules effective against ESKAPE, considering the characteristics of these pathogens. A limited number of research papers report the discovery of anti-biofilm agents effective against ESKAPE pathogens.
The authors used three different growths media to grow the four selected Antarctic bacteria – why? And why the authors selected these three media/conditions. Did the authors check many others different media/optimization conditions for the selection of the most active cell-free supernatant for each bacterial isolate? This should be discussed/highlighted in the manuscript.
The culture media used for the fermentation of Antarctic strains were synthetic, minimal and with a defined composition. For each bacterial strain, a specific growth medium was used. In detail, Pseudoalteromonas haloplanktis TAC125 and Pseudoalteromonas haloplanktis TAD2020 were grown in GG medium containing 10 g/l L-glutamate and 10 g/l D-gluconate as single carbon and nitrogen sources. Psychrobacter sp. TAE2020 was grown in GLUT medium containing 10 g/l L-glutamate as a single carbon and nitrogen source, while Pseudomonas sp. TAE6080 was cultivated in GLUC medium containing 10 g/l D-gluconate as a single carbon source. All media were complemented with a marine salt mix composed of 10 g/l NaCl, 1 g/l K2HPO4, 1 g/l NH4NO3, 200 mg/l MgSO4·7H2O, 5 mg/l FeSO4·7H2O, and 5 mg/l CaCl2·2H2O resulting in a final of pH 7.5.
The choice to use chemically defined media was to reduce the complexity of supernatant post-bacterial fermentation. For each strain, we previously set up the best media in terms of specific growth rate and low complexity, for the two Pseudoalteromonadales we formulated GG[1] while for the Psychrobacter sp.TAE2020 the most simple medium resulted to be GLUT [2], in the case of the TAE6080 the best solution was GLUC [3].
[1] F. Sannino, M. Giuliani, U. Salvatore, G.A. Apuzzo, D. de Pascale, R. Fani, M. Fondi, G. Marino, M.L. Tutino, E. Parrilli, A novel synthetic medium and expression system for subzero growth and recombinant protein production in Pseudoalteromonas haloplanktis TAC125, Appl. Microbiol. Biotechnol. (2017). https://doi.org/10.1007/s00253-016-7942-5.
[2] C. Riccardi, C. D’Angelo, M. Calvanese, A. Ricciardelli, M.L. Tutino, E. Parrilli, M. Fondi, Genome analysis of a new biosurfactants source: The Antarctic bacterium Psychrobacter sp. TAE2020, Mar. Genomics. 61 (2022) 1–5. https://doi.org/10.1016/j.margen.2021.100922.
[3] C. Riccardi, C. D’Angelo, M. Calvanese, A. Ricciardelli, A. Sellitto, G. Giurato, M.L. Tutino, A. Weisz, E. Parrilli, M. Fondi, Whole-genome sequencing of Pseudomonas sp. TAE6080, a strain capable of inhibiting Staphylococcus epidermidis biofilm, Mar. Genomics. (2021). https://doi.org/10.1016/j.margen.2021.100887.
We added the corresponding paragraph in the manuscript (lanes 202-209):
The culture media used for the fermentation of Antarctic strains were synthetic, minimal and with a defined composition. For each bacterial strain, a specific growth medium was used. The choice to use chemically defined media was to reduce the complexity of supernatant post-bacterial fermentation. For each strain, we previously set up the best media in terms of specific growth rate and low complexity, for the two Pseudoalteromonadales we formulated GG [27] while for the Psychrobacter sp.TAE2020 the most simple medium resulted to be GLUT [26], in the case of Pseudomonas sp. TAE6080 the best solution was GLUC [20].
All the activity figures are missing the controls. The culture media should be included in all the bioactivity figures as negative control. Did the authors test the culture media without any growing bacteria? The growing media could have some biological effects.
We thank the reviewer for this comment. As expected, culture media used for the growth of Antarctic bacteria influenced the biofilm formation of ESKAPE pathogens. In the manuscript we reported (Table 7) the biofilm produced by ESKAPE in each synthetic growth medium compared with the standard condition (BHI medium).
We rephrased the corresponding text to better clarify it (lanes 216-220):
As plausible, we found that biofilm growth was influenced for all ESKAPE species by the saline concentration of the medium (Table 7). To minimize this effect, as a control we grew ESKAPE strains in the same medium used for the cultures of Antarctic bacteria. Then, SNs derived from the different Antarctic bacteria were diluted 1:2 in BHI medium opportunely prepared at a concentration twice that of use.
lanes (275-281):
Anti-biofilm effect was reported as the percentage of residual biofilm after treatment in comparison with untreated bacteria grown in the same culture medium (Figg.1-6). In particular, to analyze the effect of TAE6080 SN bacteria were grown in BHI and GLUC medium; to analyze the effect of TAE2020 SN bacteria were grown in BHI and GLUT medium; and bacteria were grown in BHI and GG medium to analyze the effect of TAC125 and TAD2020 SNs In some cases, an increase in biofilm formation was highlighted after the treatment.
Round 2
Reviewer 1 Report
The authors have answered all my questions. I have no comment now.
I have no comment.
Author Response
We thank the reviewer for her/his suggestions.
Reviewer 3 Report
Unfortunately, I disagree with the authors' counterarguments. In my opinion this work has not been improved in the round of revision. The new conclusions are a "summary" of all the sections. The method of biofilm quantification is too simple.
Author Response
The conclusion section was modified in the new version following the previous suggestions of reviewer #2. We will follow the suggestions of Editor regarding this section.
As suggested by the Academic Editor, a paragraph and the corresponding references describing the advantages and disadvantages of crystal violet-based biofilm assay was added in the discussion section.
Lanes 707-718
Biofilm quantification was assessed by using crystal violet-based assay. In literature, various methods have been reported to quantify total biofilm or different components of biofilm (Wilson et al. 2017; Stiefel, et al 2016).
Different methods allow the quantification of total biomass, total amount of bacterial cells, viable cell number, and amount of extracellular polymeric substances. However, these methods are often confusedly used, leading to discrepancies and misleading results.
Crystal violet staining is a reliable method for total biomass quantification. Although crystal violet binds mainly to the biofilm matrix and does not allow distinguishing between viable or dead cells, it exhibited high reproducibility and repeatability and allows to rapidly analyze multiple samples simultaneously (Kragh et al. 2019).
In light of these reports, we think that the screening of antibiofilm activity against a large number of bacterial strains can be performed with crystal violet.
We hope that these elucidations clarify our choice about crystal violet biofilm assay.